# A Systematic Literature Review on Crop Yield Prediction with Deep Learning and Remote Sensing

**Priyanga Muruganantham \***, **Santoso Wibowo**, **Srimannarayana Grandhi**, **Nahidul Hoque Samrat and Nahina Islam**

School of Engineering and Technology, Central Queensland University, Melbourne 3000, Australia;
s.wibowo1@cqu.edu.au (S.W.); s.grandhi@cqu.edu.au (S.G.); n.smarta@cqu.edu.au (N.H.S.);
n.islam@cqu.edu.au (N.I.)
* Correspondence: p.muruganantham@cqumail.com

**Abstract:** Deep learning has emerged as a potential tool for crop yield prediction, allowing the model to automatically extract features and learn from the datasets. Meanwhile, smart farming technology enables the farmers to achieve maximum crop yield by extracting essential parameters of crop growth. This systematic literature review highlights the existing research gaps in a particular area of deep learning methodologies and guides us in analyzing the impact of vegetation indices and environmental factors on crop yield. To achieve the aims of this study, prior studies from 2012 to 2022 from various databases are collected and analyzed. The study focuses on the advantages of using deep learning in crop yield prediction, the suitable remote sensing technology based on the data acquisition requirements, and the various features that influence crop yield prediction. This study finds that Long Short-Term Memory (LSTM) and Convolutional Neural Networks (CNN) are the most widely used deep learning approaches for crop yield prediction. The commonly used remote sensing technology is satellite remote sensing technology—in particular, the use of the Moderate-Resolution Imaging Spectroradiometer (MODIS). Findings show that vegetation indices are the most used feature for crop yield prediction. However, it is also observed that the most used features in the literature do not always work for all the approaches. The main challenges of using deep learning approaches and remote sensing for crop yield prediction are how to improve the working model for better accuracy, the practical implication of the model for providing accurate information about crop yield to agriculturalists, growers, and policymakers, and the issue with the black box property.

**Keywords:** deep learning approaches; crop yield prediction; remote sensing techniques; systematic literature review

## 1. Introduction

Crop yield prediction is becoming more important because of the growing concern about food security [1–3]. Early crop yield prediction plays an important role in reducing famine by estimating the food availability for the growing world population [2]. Hunger is one of the most devastating issues in the world and increasing crop yield production is a feasible solution to overcome this problem. The World Health Organization [1] estimated that there is still an inadequate food supply for 820 million people around the world. The target for the Sustainable Development Goals of the United Nations is to eliminate hunger, accomplish food security, and encourage sustainable agriculture by 2030. The Food and Agriculture Organization (FAO) estimated that there will be a 60 per cent demand for food to supply the world population of 9.3 billion by 2050 [2]. Therefore, crop yield prediction can offer crucial information required for developing a reasonable solution to achieve the target and end hunger [1].

Crop yield is influenced by various parameters, and it is difficult to build a reliable prediction model with traditional methods. However, with advancements in computational

technology, the development and training of a novel approach for crop yield prediction have become feasible [3]. Deep learning is a significant technique that is extensively used in the agricultural domain because of its numerous data technologies and high-performance computing [4]. Deep learning is a class of machine learning that has multiple layers of neural networks capable of learning from data that are unstructured and unlabeled, whereby the learning can be supervised, semi-supervised, or unsupervised. Sarker [4] pointed out that deep learning techniques focus on learning abstract features of large datasets. To accurately predict crop yield requires primary knowledge of the association between functional attributes and interactive factors. To study such correlation requires both comprehensive datasets and high-efficiency algorithms, which can be achieved by using deep learning.

This paper conducts a systematic literature review on the application of deep learning approaches in crop yield prediction using remote sensing data. The rationale of conducting this systematic literature review is because it has the potential to highlight the existing research gaps in a particular area of deep learning methodologies and guides us in analyzing the impact of vegetation indices and environmental factors on crop growth. This systematic literature review provides a new perspective of research by investigating the advantages of using deep learning in crop yield prediction, the suitable remote sensing technology based on the data acquisition requirements, and the various features that influence crop yield prediction.

The structure of this paper is organized as follows. Section 2 provides the overview of crop yield prediction. Section 3 presents the research methodology adopted in conducting the systematic review. Section 4 presents the results and discussion, followed by the conclusions in Section 5.

## 2. Research Methods

### 2.1. Review Methodology

This systematic literature review helps us to understand the application of deep learning approaches in crop yield prediction using remote sensing data. This systematic literature review is carried out to highlight the existing research gaps in a particular area of deep learning methodologies and guide us in analyzing the impact of vegetation indices and environmental factors on crop growth. For the systematic literature review, not only are all research studies from journals, conferences, and other electronic databases assessed, but they are also integrated and presented in correspondence to the research questions mentioned in our study.

A systematic literature review is an exceptional way to evaluate theory or evidence in a specific area or to study the accuracy or validity of a specific theory [5]. The review guidelines given by Kitchenham and Charters [6] are appropriate for our systematic literature review as they provide objectivity and transparency. Based on the review guidelines, initially, the research questions are formulated. The review is undertaken in accordance with the Preferred Reporting Items for Systematic reviews and Meta-Analysis (PRISMA) statement [7]. Several databases, such as IEEE Explorer, ScienceDirect, Scopus, Google Scholar, MDPI, and Web of Science, are used for selecting the relevant research articles. These research articles are assessed and filtered based on the quality criteria. A complete checklist of PRISMA (prisma-statement.org) is used for conducting and reporting the results of the review.

### 2.2. Research Questions

The following research questions are developed to guide the systematic review:

Q1. What deep learning approaches are used for crop yield prediction?

This question helps us to analyze both the advantages and limitations of using deep learning approaches in crop yield prediction.

Q2. What remote sensing technologies are used with deep learning approaches for crop yield prediction?

With various remote sensing technologies in existence, this question helps us to understand the suitable remote sensing technology based on the data acquisition requirements for the study of crop yield prediction, such as land area and crop type.

Q3. What are the vegetation indices and environmental parameters used in crop yield prediction?

Answering this question helps us to learn about the various features that are influencing the deep learning approaches in crop yield prediction.

Q4. What are the challenges in using deep learning approaches and remote sensing for crop yield prediction?

This question helps us to understand the limitations and challenges in the existing approaches.

### 2.3. Procedure for Article Search

The approach to searching the articles is designed based on the framed research questions and the aim of the systematic literature review. Narrowing down the focus from a major concept to the central idea of the review helps in creating an effective search strategy. Using "deep learning" alone as a search string will generate a lot of published articles from various application fields that are not likely related to the aim of the review and cause the search to be complicated. Redefining the search strategy as "crop yield prediction" AND "remote sensing" AND "deep learning" can reduce the probability of deviating from the scope of the review. Initially, by using these search strings, the articles were retrieved from five databases, including IEEE Explorer, Science Direct, Scopus, Google Scholar, and MDPI. Further, to include any other relevant studies, the following keywords, namely "crop yield prediction" OR "crop yield estimation" AND "deep learning" AND "remote sensing" OR "artificial intelligence" AND "smart farming", were used to retrieve the articles from the databases. The articles from the last 10 years (2012–2022) were used for the study as deep learning approaches gained momentum after 2012 [8]. Since then, much research has been conducted on deep learning approaches.

### 2.4. Article Selection Criteria

The retrieved articles are initially selected based on aspects such as the quality of a journal, any type of remote sensing technology used for the study, and the type of deep learning approaches adopted. Analyzing the abstracts of articles helps in understanding the keywords and the selection of articles. Exclusion of irrelevant articles was carried out based on the following criteria:

- Articles that belong to the agricultural sector but that do not fall under crop yield prediction;
- Publications that include machine learning approaches for crop yield prediction;
- Publications that have no open access;
- Literature search for articles that are published before 2012;
- Articles in different languages other than English.

After applying all the exclusion criteria, a total of 51 articles are selected. Further removing the repeated articles across the selected databases, 44 articles are selected for the review. In Figure 1, we explain the process for article selection and rejection from databases for the review based on PRISMA. Table 1 shows the number of articles retrieved after selection criteria are applied and the number of articles obtained after excluding the repeated articles from the selected databases. The research questions are addressed after all the data from retrieved articles are summarized and synthesized.

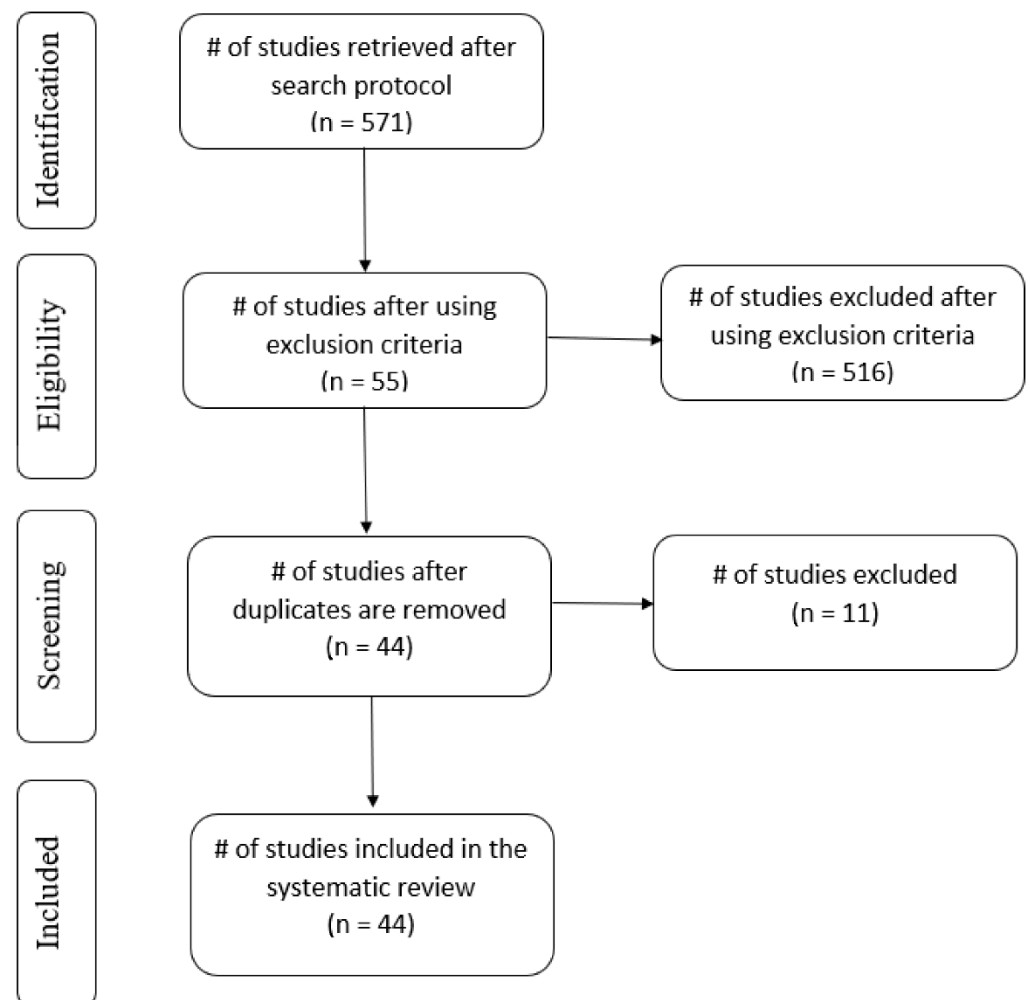

**Figure 1.** PRISMA flow chart showing the results of searches.

**Table 1.** Distribution of articles among various databases.

| Databases | # of Retrieved Articles after Search Protocol | # of Articles after Exclusion Criteria | # of Articles after Removing Repeated Articles |
|---|---|---|---|
| Scopus | 77 | 9 | 4 |
| Science Direct | 243 | 17 | 11 |
| IEEE Xplore | 20 | 7 | 7 |
| Google Scholar | 154 | 14 | 14 |
| MDPI | 13 | 4 | 4 |
| Web of Science | 64 | 4 | 4 |
| Total | 571 | 55 | 44 |

## 3. Overview of the Existing Approaches

### 3.1. Deep Learning

Developing a reliable crop yield prediction model with traditional approaches such as the static regression approach and the mechanistic approach is difficult due to their limited applicability and uncertainty [9,10]. Many studies have used machine learning approaches such as regression tree, random forest, multivariate regression, association rule mining, and artificial neural networks for crop yield prediction [9,11]. Machine learning models

treat the output, crop yield, as an implicit function of the input variables, such as weather components and soil conditions, which could be very complex [11]. Moreover, supervised learning approaches in machine learning fail to capture the nonlinear relationship between input and output variables [12]. However, the advancements in technology in recent years have made it possible to develop an advanced crop yield prediction model utilizing deep learning. Deep learning is a class of machine learning that uses hierarchical structures to link with the other layers, and the capability to analyze both unlabeled and unstructured data makes it a class apart from other traditional machine learning approaches [13]. Deep learning is broadly used in the agricultural field as it can analyze huge datasets, learn the relationships between various variables, and use nonlinear functions. These approaches can extract features for huge datasets in an unsupervised environment. When compared to traditional machine learning approaches, deep learning approaches perform better in feature extraction [13]. Since an accurate crop yield prediction relies on the factors influencing crop growth, deep learning has a strong ability to extract features from available data.

Deep neural networks have a collection of nonlinear layers that convert the untested input data into an extracted form at each layer [14]. Deep neural networks with various hidden layers are important to discover the nonlinear correlation between input and response variables [14]. Nevertheless, they are difficult to train and need recently developed hardware and optimization methodologies [15]. Thus, a rise in the number of hidden layers can be effective but it has some restrictions, which can be resolved by implementing some techniques. The vanishing gradient problem in deeper neural networks can be reduced by making use of residual skip connections for the network [15–17]. Moreover, the performance of deep learning approaches has been improved by undertaking various techniques such as stochastic gradient descent (SGD), batch normalization, and dropout. Some of the deep learning approaches are given below.

### 3.1.1. Artificial Neural Networks (ANN)

Artificial neural networks are simple neural networks that were modeled based on the human brain's neural structure [14]. The neural network consists of nodes, which are connected to each other, where the neurons are grouped into layers. The network has three layers: the input layer, the hidden layer, and the out layer. The inputs are received by the input layer of neurons; the hidden layer with interconnected neurons performs the function and then provides the output to the output layer [16]. Moreover, to initiate the process, initial weights are assigned randomly.

### 3.1.2. Deep Neural Networks (DNN)

A DNN is a special kind of feed-forward neural network with many hidden layers that are fully connected. Generally, activation functions such as ReLU and loss functions such as L2 regularization and mean squared error are used with the hidden layers [13].

### 3.1.3. Bayesian Neural Networks (BNN)

BNN uses a neural network with Bayesian inference and probability distributions are used as weights in BNN. Using a Bayesian neural network can prevent the problem of overfitting without necessary validation data to evaluate the regularization parameter [18]. For better accuracy, training a BNN with a large dataset can be helpful.

### 3.1.4. Convolution Neural Network (CNN)

Compared to conventional neural network approaches, a CNN includes layers such as convolution layers, pooling layers, and fully connected layers, which helps in efficiently finding salient features within the data. The convolution layer consists of a convolution operation and activation function, which perform feature extraction [19]. Convolution operation includes a filter and feature map. A filter is a group of weights applied across the input and a feature map is the corresponding output for a given filter. Moreover, a pooling operation is used to perform down-sampling as it helps to detect features effectively [19].

The outputs are then passed through a nonlinear activation function as it generates nonlinearity into the output. Fully connected (FC) layers are used after convolution layers; the network has the capacity to learn the mapping between the feature and the target by increasing the FC layers [20].

### 3.1.5. 2D-CNN and 3D-CNN

A 2D-CNN is called a spatial method whereas 3D-CNN is called a spatio-temporal method [16]. In a 2D-CNN, the input data are considered as the spatial–spectral volume, where the kernel slides along the two spatial dimensions that are across width and height. In a 3D-CNN, to the two spatial dimensions, a temporal dimension is also added. A 3D-CNN uses three-dimensional kernels, which slide along width, height, and depth and help in generating a 3D feature map [21]. The 3D-CNN approach is developed by implementing 3D convolutional layers [21].

### 3.1.6. Faster R-CNN

The region-based convolutional neural network (R-CNN) is predominantly used for object localization and object detection [22]. There are four different kinds of R-CNN; they are R-CNN, Fast R-CNN, Faster R-CNN, and Mask R-CNN. The difference in pooling methods and region proposal methods makes the R-CNNs different and their process faster.

### 3.1.7. Long Short-Term Memory (LSTM)

LSTM is a special kind of recurrent neural network (RNN) that can learn time-dependent information with an appropriate gradient-based algorithm. The LSTM comprises a chain structure, which starts with an input layer, one or more LSTM layers, and the output layer. To control the cell state and output, the LSTM uses three gates, namely the input gate, forget gate, and output gate. These gates are more likely as neural network layers, which can control the information transfer [14]. Each cell in LSTM layers has three gates; the input gate decides which information needs to be retained, the forget gate determines the amount of previous information that must be forgotten and the amount of current input that needs to be reserved, the output gate uses both the current input and previous output to decide the final output [23]. Tian et al. [24] proposed the ALSTM model, which had six layers, namely one input layer, one LSTM layer, one attention mechanism layer, two dropout layers, and one output layer.

### 3.2. Remote Sensing for Data Acquisition

Crop yield can vary according to environmental factors, climatic conditions, disease, and other parameters. The crop growth during different stages is influenced by these above-mentioned factors and this reflects on crop production [10]. Monitoring of environmental factors, other parameters, and crop growth can be carried out using various instruments and methodologies, such as ground observation, remote sensing, global positioning systems, and on-field surveying. In ground observation and other traditional methods, it is challenging to personally acquire data for a large area and the result will be less accurate and unreliable [10]. To counteract this limitation, nowadays, remote sensing is increasingly utilized for crop monitoring.

Remote sensing techniques provide details about the status of crops at various growth levels through the use of spectral signatures, and all at a minimum cost when compared to extensive on-field surveying. Remote sensing technology is the acquisition and analysis of information about the world and its objects by an instrument placed in the atmosphere or a satellite, without any physical contact [25]. Remote sensing has the ability to produce an adequate number of data when compared with other data acquiring techniques such as field surveying [26]. It is the process of monitoring and recognizing places on Earth by measuring the emitted and reflected radiation with the help of sensors [27]. Data acquired using remote sensing have several applications in agriculture, which include crop

type classification, crop yield prediction, soil property detection, crop health monitoring, weather data assessment, and soil moisture retrieval [28].

One of the most important reasons to use optical remote sensing for acquiring crop information is due to the computation of vegetation indices. Combinations of spectral measurements at various wavelengths are known as spectral indices. They are employed to derive vegetation phenology and calculate biophysical parameters [29]. Among various spectral indices, vegetation indices are the desired indices significantly used in crop yield prediction. Crops in healthy condition are indicated by strong absorption and reflectance of red and near-infrared bands [30]. The strong difference in the intensities of the absorption and reflectance of red and near-infrared bands can be integrated into various quantitative indices of the vegetation environment. These linear or nonlinear combinational operations are known as vegetation indices (VI) [31,32]. Some of the VI are the normalized difference vegetation index (NDVI), green vegetation index (GVI), chlorophyll absorption ratio index (CARI), and many others.

*3.3. Impact of Vegetation Indices and Environmental Factors*

Vegetation indices are formulated in which the sensitivity to the vegetation characteristics is maximized while the factors such as soil background reflectance and directional or atmospheric effects are minimized. Most of the vegetation indices use information involving the red and near-infrared (NIR) canopy reflectances or radiances [33]. A satellite with a multispectral sensor and several bands covers the visible, near-infrared, and short-wave infrared wavelength regions, which provides numerous vegetation indices.

Different types of vegetation indices are designed by various researchers and are extensively utilized in several research areas. Even though there is some variation in these proposed indices, all these designed indices are sensitive to biochemical attributes and biophysical parameters such as leaf angle distribution function, leaf chemical contents, fraction of absorbed photosynthetically active radiation, biomass, fraction of green coverage, and leaf area index (LAI). Due to the strong correlation between vegetation indices and biophysical parameters, they are widely used to determine the nutritional level of plants, mostly relative to nitrogen [34,35], to classify vegetation and to schedule crop management. However, the phenological stage of evaluation and the type of indices utilized influence their accuracy. Zhao et al. [35] developed a function that established a relationship between the crop coefficient (Kc) for irrigation management and the vegetation index, and it was used in water conservation. Other significant areas where these indices are used are the estimation of crop yield, protein content, biomass, weed management, and fertilizer management [36–38]. Some of the most commonly used indices are the normalized difference vegetation index (NDVI), green vegetation index (GVI), enhanced vegetation index (EVI), chlorophyll absorption ratio index (CARI), and many others.

Various studies have investigated how the correlation between remotely sensed data and crop yield differs as a function of time in the course of the growing season [39,40]. The studies have indicated that the relationship between vegetation indices and crop yield varies during the crop growth cycle [41–44]. Moreover, the relationship between vegetation indices and crop yield is not consistent in every growth stage. For example, the suitable phenological growth stages for wheat to obtain spatial yield data from satellite remote sensing are stem elongation, heading, and the development of fruit until early ripening [45–47]. Similarly, Ali et al. [48] proposed that the appropriate crop growth stage to study the correlation between vegetation index and crop yield and to estimate crop yield and biomass for oat grain was the appearance stage of the leaf health and kernel watery ripe stage. Most previous studies have performed correlation analyses between vegetation indices, soil data, and yield data and were mostly carried out for particular crop types, specific years, and a restricted number of vegetation indices [48–51]. The ultimate goal in correlation analysis between vegetation indices and crop yield data is to develop an optimal crop yield prediction model [52–54].

For developing a workable crop yield prediction model, it is important to determine the appropriate vegetation indices and environmental factors [55,56]. You et al. [57] predicted corn yield using the greenness index with 90% accuracy. Fernandes et al. [58] noticed that the crop yield is influenced by vegetation indices selection. Their study on maize yield prediction showed that NDVIre, NDVI, and GNDVI performed well for field variability. Haghverdi et al. [59] observed that crop yield prediction with NDVIre was more effective when compared to NDVI and GNDVI. Wang et al. [60] estimated the corn yield by combining vegetation indices such as the normalized vegetation index (NVDI) and Absorbed Photosynthetically Active Radiation (APAR) with environmental factors including canopy surface temperature and water stress index [56]. Further, other features, such as humidity, nutrients, and soil information, are also used in crop yield prediction. As so many features are already used in crop yield prediction, there is less investigation related to finding specific features majorly impacting crop yield prediction. Hence, detailed research is essential to achieve a better overview of these variables and factors influencing crop yield prediction.

## 4. Results and Discussion

The articles selected for the review are analyzed and summarized. Figure 2 shows the number of articles published between 2012 and 2022. For the years 2012–2014, the articles that we retrieved did not satisfy the condition of using both deep learning and remote sensing to predict crop yield. It is evident that the study of crop yield prediction using remote sensing has increased in recent years. Table 2 provides a detailed review of the type of remote sensing used for the study, data and parameters used in the study, and the model.

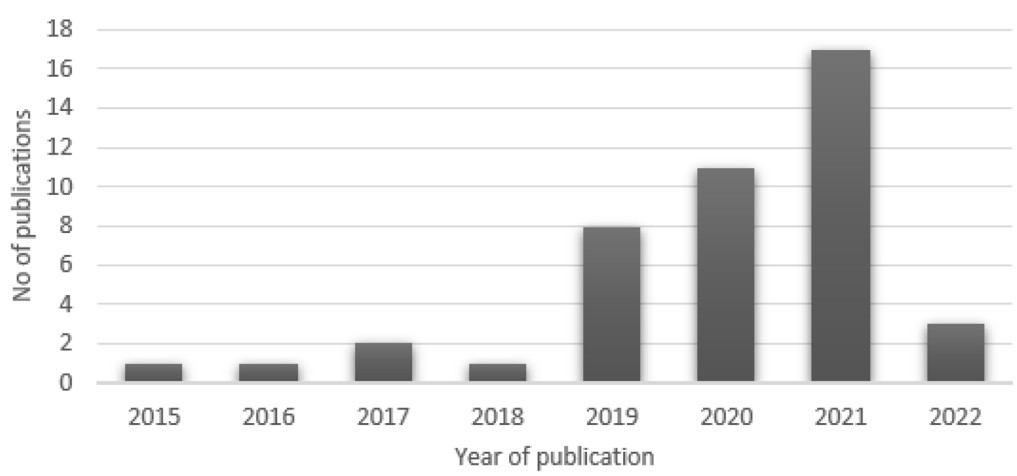

**Figure 2.** Distribution of articles between 2012 and 2022.

**Table 2.** Summary of articles between 2012 and 2022.

| Type of Remote Sensing | Model | Data and Features Used in Study | Authors |
|---|---|---|---|
| AVHRR | Bayesian neural networks (BNN) | Crop yield data | Johnson et al. [16] |
| | LSTM | Yield data, surface reflectance, NDVI, air temperature, precipitation, air pressure, humidity | Wang et al. [60] |

**Table 2.** *Cont.*

| Type of Remote Sensing | Model | Data and Features Used in Study | Authors |
|---|---|---|---|
| ERA5 reanalysis | LSTM | Growing Degree Days (GDD), plant–harvest cycle, average yield data, maximum temperature, minimum temperature | Cunha and Silva [61] |
| Landsat 8 | ANN | NDVI, SR, NIR, GNDVI, GI, WI, and SBI | Haghverdi et al. [57] |
| MODIS | 3D-CNN | Surface reflectance, land surface temperature, and land cover type | Gavahi et al. [62], Qiao et al. [63], Abbaszadeh et al. [64] |
| | CNN | NDVI, NDWI, NIR, precipitation, minimum, mean and maximum air temperatures | Kuwata and Shibasaki [56], Mu et al. [65], Terliksiz and Altýlar [66], Wolanin et al. [67], Cao et al. [68] |
| | CNN-LSTM | Surface reflectance, land surface temperature | You et al. [57], Sun et al. [69]. Sharma et al. [70], Ghazaryan et al. [71], Gastli et al. [72], Jeong et al. [73] |
| | DNN | NDVI, Absorbed Photosynthetically Active Radiation (APAR), land surface temperature | Dang et al. [74], Gao et al. [75] |
| | Deep forward neural network (DFNN) | Yield data, surface reflectance, land surface temperature, cropland data layers | Khaki et al. [76] |
| | LSTM | NDVI, EVI, land surface temperature | Tian et al. [23], Tian et al. [24], Kaneko. et al. [77], Jiang et al. [78], Ma et al. [79], Zhang et al. [80], Xie et al. [81] |
| | Neural networks ensemble | NDVI, Red, SR, NIR, GNDVI, GI, WI, and SBI | Fernandes et al. [58] |
| Sentinel-2 | 3D-CNN | Crop yield, rice crop mask, B02-B08, B8A, B11, B12 and NDVI, climate data | Fernandez-Beltran et al. [21] |
| | DNN | Precipitation, temperature, NIR, and SWIR | Jin et al. [82], Engen et al. [83] |
| | LSTM | Minimum and maximum temperature, integrated solar radiation, cumulative precipitation, soil texture, soil chemical parameters, hydrological properties | Xie et al. [84] |
| UAV | CNN | EVI, GRVI, GNDVI, MSAVI, OSAVI, NDVI, SAVI, WDRVI | Nevavuori et al. [85], Yang et al. [86], Yang et al. [87], Yang et al. [88] |
| | CNN-LSTM | RGB images, thermal time, crop yield data, cumulative temperature | Nevavuori et al. [85] |
| | DNN | NDVI, GNDVI, EVI, EVI2, WDRVI, SIPI, NRVI, VARI, TVI, OSAVI, MCARI, TCARI, NDWI, NDRE, RECI, GLCM | Sagan et al. [89] |
| | Faster R-CNN | Weather images | Chen et al. [22] |

**Table 2.** *Cont.*

| Type of Remote Sensing | Model | Data and Features Used in Study | Authors |
|---|---|---|---|
| UAV | Multimodal data fusion | Surface temperature, air temperature, humidity, normalized relative canopy temperature (NRCT), Vegetation Fraction (VF) | Maimaitijiang et al. [90] |
| | Spectral deep neural network (sp-DNN) | Crop yield, harvested yield, multispectral images, NIR, NDVI, NDVI-RE, NDRE, ENVI, CCCI, GNDVI, GLI, and OSAVI | Danilevica et al. [91] |

Based on the data analysis, the following research questions can be addressed:

RQ1 —Approaches used in literature discussion:

For the first research question (RQ1), the deep learning approaches used for crop yield prediction are summarized in Table 2. Some unique approaches used are Neuroevolution of Augmenting Topologies (NEAT) and YieldNet. Table 2 shows that LSTM and CNN are the most used deep learning approaches for crop yield prediction. Apart from using CNN and LSTM approaches, these approaches are widely used together with CNN-LSTM, multi-level deep learning approaches with multiple levels, and when using multimodal fusion approaches. It is found that simple neural networks are least used for crop yield prediction when using remote sensing data. The use of Neuroevolution of Augmenting Topologies (NEAT) in ANN, Caffe for implementing CNN, and Faster R-CNN are some promising aspects of deep learning. Most of the approaches included data-pre-processing, and in some approaches, the Gaussian process was used along with CNN and LSTM. Figure 3 shows the distribution of crop yield prediction articles based on deep learning approaches.

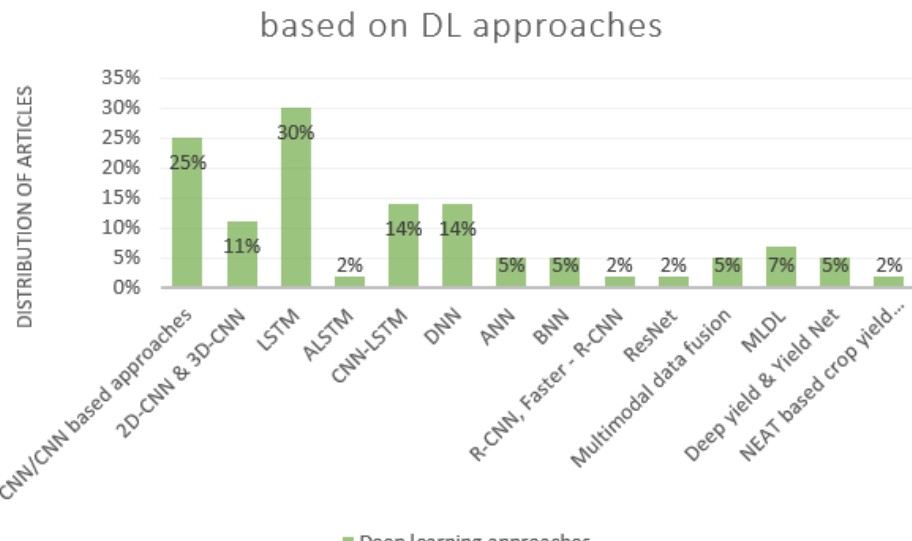

**Figure 3.** Distribution of articles based on deep learning approaches.

Table 2 shows that one of the most frequently used deep learning approaches for crop yield prediction is the convolutional neural network (CNN). It is widely used because of its special ability to find the important features within the data. Nevavuori et al. [20] studied crop yield prediction with a unique profile of temperature and photoperiod, developing a region-specific deep learning model. Terliksiz and Altýlar [66] explained that choosing the right data frame for 3D-CNN is significant for crop yield prediction. Wolanin et al. [67]

analyzed the influence of reducing the input time series in 25 days on model performance, and the CNN model with 10 variables had a better prediction for days during the flowering stage of wheat. Nevavuori et al. [85] developed 3D-CNN, ConvLSTM, and CNN-LSTM for modeling crop yield. Among them, a model with five 3D-CNN layers had good performance. Yang et al. [86] studied crop yield prediction during the ripening stage, in which the VI-based model had poor performance as the maximum greenness stage had an influence on the VIs, but the CNN with RGB and multispectral images performed well even in the fully ripened stage. Yang et al. [87] proposed a CNN approach to study corn yield, in which the CNN using spectral and color image features performed well when compared to the 1D-CNN and 2D-CNN using spectral and spatial features.

LSTM was also used extensively in crop yield prediction, and its ability to learn time-dependent information makes it different from the other deep learning approaches. Wang et al. [60] studied the model performance accuracy by using a different combination of remote sensing and meteorological data and soil data. The addition of soil data was an advantage for the model in acquiring the spatial variability of yield. Tian et al. [23] used different combinations of input data, such as a combination of vegetation temperature condition index (VTCI) and remote sensing data and a combination of VTCI, meteorological data, and remote sensing data, to identify the best-performing wheat yield estimation model. The authors pointed out that the incorporation of VTCI with remote sensing and meteorological data can acquire information about various influences on crop growth. Cunha and Silva [61] studied crop yield prediction using LSTM and performed thirty different train–test cycles to assess the performance. Their approach was to use weather and soil data rather than satellite data as NDVI-based crop yield prediction is achievable only after a certain plant growth stage, whereas weather-based prediction is feasible before the planting period. Kaneko et al. [77] studied crop yield prediction in a group of countries using the LSTM approach. LSTM with Gaussian processing performed well for random splits when compared to the chronological splits. Jiang et al. [78] proposed an LSTM approach; prediction accuracy was good in extreme weather conditions compared to LASSO and RF. Moreover, LSTM performed well in learning temporal features when compared to ML approaches such as RF and LASSO. Zhang et al. [80] also found that the combination of input data performs well for crop yield prediction and estimated a suitable VI for crop yield prediction. The authors trained the model with six VIs and two climatic indices individually to choose the optimal CI, and then found the optimal combination for crop yield prediction from the following optimal VI and environmental indices (EI) combination, optimal CI and EI combination, optimal VI and optimal CI combination, and optimal VI, optimal CI, and EI combination. Incorporating all these indices helped to capture important variations in maize yield prediction.

Tian et al. [24] applied a stepwise sensitive analysis method to investigate the significance of each input variable in wheat yield prediction, which can help in understanding the ALSTM algorithm. ALSTM has two parts: one is the LSTM network and the other is the attention mechanism. The ALSTM's performance was better when compared to LSTM. Moreover, ALSTM has good generalization ability and performed well even though the sampling sites had varied farming systems.

The CNN and LSTM approaches are combined and are studied as CNN-LSTM approaches. Sharma et al. [70] studied crop yield estimation using the CNN-LSTM approach. They studied the influence of contextual information such as water bodies, farmlands, and urban landscape in crop yield prediction by comparing it with another prediction model using CNN-LSTM without these factors. The CNN-LSTM along with contextual information performed well, thus improving the yield estimation. Gastli et al. [72] used CNN-LSTM with Gaussian Process (GP) to predict crop yield in California and compared the approach with CNN and LSTM. It was found that CNN-LSTM with GP performed better on data with and without moisture and histograms. When compared to CNN and LSTM, the CNN-LSTM aims to capture features effectively. Sun et al. [92] used the CNN-LSTM approach to study the end-of-season crop yield prediction and compared its

performance with the CNN and LSTM approaches. The CNN-LSTM-based end-of-season model performed well when compared to both CNN and LSTM for five years. Compared to environmental features, the MODIS surface reflectance had a significant influence on the CNN-LSTM-based model.

DNN was also mostly used in crop yield prediction, either individually or in multimodal fusion. Cao et al. [68] compared DNN with ML approaches such as SVM and random forest to find the best winter wheat yield prediction model. Jin et al. [82] studied biomass estimation using the DNN approach. Initially, the DNN performed well for biomass estimation with 15 vegetation indices; however, the biomass estimation accuracy was improved when LAI was combined with 15 vegetation indices. Ma et al. [79] compared BNN with ML approaches, and BNN performed well in both end-of-season and within-season crop yield prediction.

Chen et al. [22] used a region-based convolutional neural network (R-CNN) as this reduces the number of proposed regions generated while ensuring precise object detection. Moreover, the authors compared the R-CNN, Fast R-CNN, and Faster R-CNN performance for detecting strawberry flower and fruit. The Faster R-CNN showed better performance with the lowest training time and had the fastest detection rate.

Multimodal data fusion, a fundamental method of multimodal data mining, aims to integrate the data of different distributions, sources, and types into a global space in which both intermodality and cross-modality can be represented uniformly [93–96]. Gavahi et al. [62] proposed a deep yield approach by combining 3D-CNN and Conv-LSTM networks together. Maimaitijiang et al. [90] used feature fusion at the input level and intermediate level within a DNN framework to predict crop yield. The proposed multimodal deep learning performed well, and also when considering prediction accuracy, spatial adaptivity, and robustness, the overall performance of the intermediate-level feature fusion DNN framework was comparatively higher than the input-level feature fusion DNN framework. Similarly, Danilevicz et al. [91] used multimodal deep learning by incorporating tab-DNN, sp-DNN, a fusion module with two linear layers, and ReLU. For the fusion module input, the authors concatenated the weights from the last layer of tab-DNN and sp-DNN. The approach performed well for early crop yield prediction. However, in MLDL/ensembling techniques, several networks are stacked in levels and the model uses the features extracted by several networks. Moreover, Sun et al. [92] proposed the MLDL, which is based on combining models together to form a single network and stacking it in different levels with convolution layers, pooling layers, and fully connected layers.

A deep residual network or ResNet is a specific type of neural network that was introduced to avoid the vanishing/gradient problem. ResNet can assist in training up to a thousand layers of deep networks. The core element of ResNet is the residual blocks, and the skip connections in ResNet help to skip any layer that affects the performance of the approach using regularization [77]. The YieldNet framework was proposed by Khaki et al. [97], where they used a transfer learning methodology to share weights for the backbone feature extractor. Based on this approach, the authors predicted the yield of corn and soy simultaneously. Moreover, when compared to YieldNet, other approaches such as 3D-CNN and DFNN required more parameters for prediction and longer training time. Barbosa et al. [81] proposed an approach called NEAT, where they used a genetic algorithm called Neuroevolution of Augmenting Topologies (NEAT) on the proposed dataset. NEAT is used to adjust both topology and weight parameters in the development of the artificial neural network for crop yield prediction.

It can be seen that CNN approaches have performed well in dealing with data with similar features. Yang et al. [87] used a CNN to predict crop yield at the ripening stage. The CNN approach performed well when both the test set and training set had the same phenological stage. The CNN-LSTM model performs better when predicting crop yield with a large dataset [60,68,78,93]. From Table 2, based on the evaluation metrics, the CNN- and LSTM-based approaches have better performance when compared to ANN and DNN approaches.

From the study, we have observed that multiple software programs were used to process the data and various techniques were adopted to overcome overfitting in deep learning approaches. Fernandes et al. [58] used TIMESAT to extract a set of metrics from the NDVI growing curves and they only considered the unmixed sugarcane pixels while extracting metrics for each municipality in Brazil using an Interactive Data Language (IDL) technique. The wrapper with sequential backward elimination technique with ANN for feature extraction was carried out as input in a stacking ensemble neural network model. In order to overcome overfitting, the authors divided the datasets into three parts (70% for training, 10% for validation, and 20% for testing). Haghervedi et al. [59] calculated vegetation indices and implemented the ANN approach using Neurosolution V 7.1.1.1 software. The authors used the technique of stopping the training when the mean square error of the cross-validation subset started to increase or showed no improvement after 100 iterations to avoid overfitting. For UAV images, Chen et al. [22] incorporated the idea of using orthoimages to avoid the distortion of photos, and all the images were labeled manually using the labeling program developed by the Computer Science and Artificial Intelligence Laboratory (MIT, MA, USA). Kaneko et al. [77] used the surface reflectance data (all seven bands of the MOD09A1.006) and temperature data (two bands of the MOD11A2.006) of MODIS to generate features such as NDVI and EVI. The authors employed the histogram dimensionality reduction approach with LSTM to counteract overfitting as the quantity of label data might be sparse. Nevavuori et al. [20] processed the irregular set of data points obtained from the yield measurement devices as rasters of field-wise yield. FarmWorks software was used for filtering and the generation of raterizable vector files. The authors extracted field-wise image data (UAV images) and yield values using the sliding window technique to obtain geolocational matching pairs of inputs and targets. To overcome the problem of overfitting, two distinct regularization strategies with CNN, such as weight decay and early stopping, are used. The GEE-based transformation was used by Sun et al. [69] to transform an entire image collection composed of remote sensing data into a 32-bin normalized histogram by country level. The authors employed the early stopping regularization technique to reduce the generalization error in the CNN-LSTM model. The pooling layer with convolution layer were used to extract features [62]. Sharma et al. [70] did not include a pooling layer in the CNN-LSTM approach because of strided convolutions. Regularization strategies such as early stopping, $L^2$ penalty, and the dropout technique were used in CNN- and LSTM-based approaches to overcome the problem of overfitting.

Some of the software and techniques used for calculating VIs, processing, and pre-processing the data are summarized below:

- TIMESAT;
- Neurosolution version 7.1.1.1;
- Use of Orthoimages;
- The labeling program developed by the Computer Science and Artificial Intelligence Laboratory (MIT, Massachusetts, USA);
- FarmWorks;
- Google Earth Engine-based tensor generation;
- Orthomosaic map generation (RGB images)—Agisoft PhotoScan Professional 1.2.5;
- Orthomosaic reflectance map (multispectral images)—Pix4Dmapper 4.0;
- Georeferencing—Esri ArcGIS 10.3;
- The clipping process—"ExtractByMask" function from Arcpy Python module;
- Layer stacking of images;
- Mosaic and orthorectify, lens distortion, and vignetting issue correction (UAV RGB images)—Pix4Dmapper software;
- Drawing of shapefile—plotshpcreate of R library;
- Conversion of Geotiff plots to Numpy arrays—Python script;
- Dimension transform technique—irregular shaped images;
- MODIS products;

- Image pre-processing—Spectronon software (version 2.134; Resonon, Inc., Bozeman, Montana);
- Spectral data denoising—wavelet transform technique.

The overall result of this study indicates that the use of deep learning approaches for crop yield estimation has increased over the years, and the number of articles retrieved from our initial search strategy before applying exclusion criteria indicates the considerable amount of research carried out on crop yield prediction. However, the number of articles using remote sensing and deep learning approaches for crop yield prediction is still relatively low. Even after selecting possible potential articles for this review, there is a possibility that some valuable articles might have been missed. However, the effectiveness of this systematic literature review can be validated considering the process adopted to conduct this review and the search string used to retrieve the maximum number of articles related to this review.

RQ2 —Remote sensing used with deep learning:

Table 2 shows that the most commonly used remote sensing technology was satellite remote sensing technology—in particular, MODIS. The data, such as surface reflectance, land surface temperature, and annual land cover, were obtained from MODIS. The other satellite remote sensing technologies used along with deep learning were Landsat 8 and Landsat 7, Sentinel-2, WorldView-3, and PlanetScope. It can be observed that satellite remote sensing has the ability to produce an adequate amount of data. The satellite remote sensing is a reliable source of data acquisition as it is easily available and inexpensive when compared to other remote sensing technologies. The multispectral satellites can offer huge amounts of information on crop yield using high spatial and temporal resolution [57]. The raw bands from the remote sensing technology help to derive the vegetation indices and the extracted features are used with DL for crop yield prediction. The correlation between these features and the crop yield was studied in the articles to understand the efficiency of these features in crop yield prediction. Sometimes, the use of all the derived vegetation indices does not help significantly in generating better performance of the approach. It is observed that the use of remote sensing technology for data acquisition is based on the requirements of the study. For instance, satellite images are more useful when dealing with a large area of land, such as a state or district. Figure 4 depicts the distribution of articles using satellite data across years 2015–2022. It can be seen that the majority of the articles on the use of satellite data with deep learning were published in 2015 and 2020, with 39% and 25%, respectively.

RQ3 —Features used in crop yield prediction

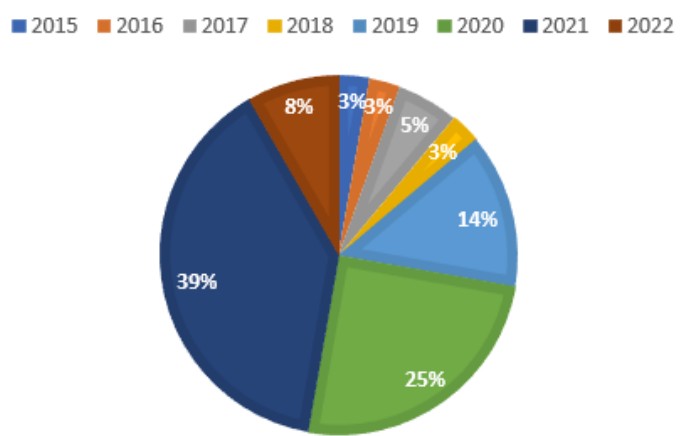

**Figure 4.** Distribution of articles on the use of satellite data with deep learning between 2015 and 2022.

The features used for crop yield prediction are provided in Table 3. Following this method helped in viewing the different types of features respective to their selected crop and the selected approach used for crop yield prediction. Crop yield data were uniquely used in all the articles as it is a key feature in crop yield prediction. Apart from vegetation indices obtained from remote sensing technologies, other features such as temperature and precipitation were grouped under meteorological data in the articles retrieved. Similarly, features such as soil type, clay content, silt content, sand content, bulk density, and coarse fragments were grouped under soil data. Moreover, features such as growing degree days and killing degree days were used in country-level model performance. Features such as pH value, Total Potential Evapotranspiration (PET), biomass, and cropland information were also used in the studies.

**Table 3.** Features that influence crop yield.

| | |
|---|---|
| Vegetation Indices | NDVI—Normalized Difference Vegetation Index, EVI—Enhanced Vegetation Index, GCI—Green Chlorophyll Index, NDWI—Normalized Difference Water Index, NIR—Near-Infrared, MODIS Surface Reflectance, MODIS Land Surface Temperature, GNDVI—Green Normalized Difference Vegetation Index, EVI2—Two-Band Enhanced Vegetation Index, WDRVI—Wide Dynamic Range Vegetation Index, SIPI—Structure Insensitive Pigment Index, NRVI—Normalized Ration Vegetation Index, VARI—Visible Atmospherically Resistant Index, TVI—Triangular Vegetation Index, OSAVI—Optimized Soil Adjusted Vegetation Index, MCARI—Modified Chlorophyll Absorption Ratio Index, TCARI—Transformed Chlorophyll Absorption Reflectance Index, NDRE—Normalized Difference Red-Edge Index, RECI—Red-Edge Chlorophyll Index, GLCM—Gray-Level Co-Occurrence Matrix, GI—Greenness Index, WI—Wetness Index, Red Band, SBI—Soil Brightness Index, SAVI—Soil-Adjusted Vegetation Index, MSAVI—Modified Soil-Adjusted Vegetation Index, SIF—Solar-Induced Chlorophyll Fluorescence, APAR—Absorbed Photosynthetically Active Radiation, PCI—Precipitation Condition Index, VHI—Vegetation Health Index, PAR—Photosynthetically Active Radiation, TVDI—Temperature Vegetation Dryness Index, VSWI—Vegetation Supply Water Index, PDI—Perpendicular Drought Index, RZSM—Root Zone Soil Moisture, NDMI—Normalized Difference Moisture Index, LAI—Leaf Area Index, ET—Total Evapotranspiration, LE—Average Latent Heat Flux, PET—Total Potential Evapotranspiration, PLE—Average Potential Latent Heat Flux, GPP—Gross Primary Productivity, PsnNet—Net Photosynthesis, CCCI—Canopy Chlorophyll Content Index, GLI—Green Leaf Index, NDVI-RE—Normalized Difference Vegetation Index Red Edge, NDRE-R—Normalized Difference Red Edge Red, VTCI—Vegetation temperature Condition Index, NRCT—Normalized Relative Canopy Temperature, VF—Vegetation Fraction, Sentinel—2—Bands B02 to B08, B8A, B11, B12, RDVI—Renormalized Difference Vegetation index, MTVI1—Modified Triangular Vegetation Index, TBWI—Three-Band Water Index, WDRVI—Wide Dynamic Range Vegetation Index, NDII—Normalized Difference Infrared Index, DCNI—Canopy Nitrogen Index |
| Meteorological Data/Weather Conditions | Precipitation, minimum temperature, mean temperature, maximum air temperature, temperature, weather, accumulated precipitation, cumulative temperature, average precipitation, average temperature, GDD—Growing Degree Days, KDD—Killing Degree Days, FDD—Frozen Degree Days, Surface Downward Shortwave Radiation Flux (SWdown), Water Vapor Pressure Deficit (VPD), air pressure, air-specific humidity, surface downward longwave radiation, wind speed, evapotranspiration, water stress indicator |
| Crop Yield Information (Excluding Crop Yield Data) | Growing phase as percentage of total thermal time, start of crop season, end of crop season, length of crop season, harvest cycle, country-level yield, field-level yield, biomass measure, fresh grain yield, dry grain yield, wheat crop fraction |
| Images | RGB images, hyperspectral images, moisture images |
| Soil Data | Clay content mass fraction, sand content mass fraction, water content, pH, bulk density, carbon content, silt content, coarse fragments, cation exchange capacity, pH in $H_2O$, pH in KCL, Soil Available Water Holding Capacity (AWC), particle size distribution, total nitrogen |
| Others | Annual land cover, plant growth stage, micro-topographic fields, plant height, crop planting areas, thermal time, solar radiation, crop land data |

Zhang et al. [80] compared the satellite-derived cumulative precipitation, standardized precipitation index, and land surface temperature (MODIS daily LST) performance with the ground-observed climate metrics, and the satellite-derived climate metrics performed well.

The authors state that the use of MODIS Land Surface Temperature over air temperature is effective as the LST-based growing, killing, and frozen degree days were able to explain most of the inter-annual variation in yield. Moreover, LST is reactive to the surface–energy balance, leaf canopy temperatures, and plant moisture, which makes it advantageous in crop yield prediction. However, the performance of satellite-based precipitation was comparatively lower than the temperature-based metrics. Moreover, in some articles, RGB images have outperformed spectral images. Nevavuori et al. [85] established that RGB images' performance was better than that of NDVI images. Maimaitijiang et al. [90] found that the RGB-based information had comparatively better prediction accuracy than the spectral features.

You et al. [57] observed that the growth-related infrared band and the short-wave infrared band were significant during crop growing months, whereas land surface temperature bands had importance in earlier crop growing months. Fernandes et al. [58] studied the multi-temporal component of Sentinel-2 images as an important element in the improvement of rice crop classification accuracy. The use of auxiliary climate and soil data helped in crop yield estimation only for small patch sizes. Haghverdi et al. [59] noted that the Greenness Index and Wetness Index were the most effective features compared to the normalized difference vegetation index (NDVI), red band, simple ratio, near-infrared band, green normalized difference vegetation index, and soil brightness index. Barbosa et al. [94] observed that LAI was the most effective feature among the Leaf Area Index (LAI), tree height, crown diameter, and the RGB values based on seven different regression algorithms and feature selection. The features that influence crop yield are grouped and summarized in Table 3 below. In Figure 5, we present the distribution features across articles. It can be seen that vegetation indices are the most commonly used feature for crop yield prediction. However, it is also observed that the most used features in the literature do not always work for all the approaches.

RQ4 —Challenges in using deep learning approaches and remote sensing for crop yield prediction

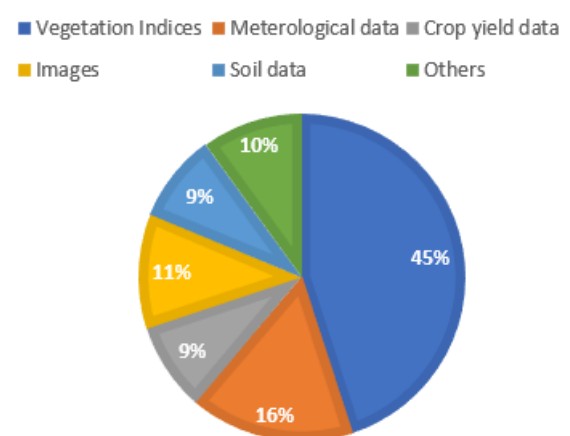

**Figure 5.** Distribution of features across articles.

The challenges described here are based on the limitations mentioned in the retrieved articles. The challenge is in how to improve the working model for better accuracy by using a larger dataset to train, testing the model for better performance, and learning the micro-topological changes affecting the crop yield prediction [98]. Another challenge is to investigate the significant practical implication of the model for providing accurate information about crop yield to agriculturalists, growers, and policymakers. The black box property was also one of the major challenges faced when using deep learning approaches

as it is difficult to understand the working behind the neural networks, and understanding this may help to develop a better model. Khaki et al. [99] have trained DNN and CNN-RNN using the backpropagation method to perform feature selection for reducing the black box nature of the model. The authors used this feature selection method to effectively evaluate the individual effects of weather components, soil conditions, and management variables, as well as the time period. To improve the traceability and interpretability of the LSTM model, recurrent network structures such as the Gated Recurrent Unit and basic RNN, and advanced analytical tools such as the attention network, can be further explored in the model [67]. Moreover, as the influence of the crop growth phase/phenological stage in crop yield prediction is significant, any change in the climatic conditions during crop growth stages can drastically affect the plant growth, which eventually leads to variation in crop yield estimation. Nevavuori et al. [85] observed the significance of the crop growth phase in crop yield prediction using weather information at a city scale; however, the use of local temperatures and other climatic variables from local weather stations can improve the possibility to learn the accuracy of change in crop growth stages.

The ANN approach was not efficient to capture the variability in micro-topography, which affected the prediction and had large model errors for fields with high yield variability [75]. The challenge with deep learning approaches is that large amounts of data are required to achieve good accuracy, and also the complexity of deep learning approaches increases the time complexity of the algorithm [76]. Terliksiz and Altýlar [66] observed that the results can be affected when data frames with different cropland coverage are fed to the 3D-CNN. Ma et al. [79] mentioned that incorporating the Bayesian deep learning approach with another deep learning approach such as LSTM, which has the potential to handle time-series features, can help in improving the model. Apart from the features mentioned in the previous section, the inclusion of other factors such as phenological stages, crop varieties, water availability, and others can help in the development of a potential model [69,75]. Moreover, the challenge lies in developing an approach with potential variables that is likely to perform well for all crops or specific kinds of crops.

## 5. Conclusions

This paper has presented a systematic literature review on the application of deep learning approaches for crop yield prediction using remote sensing data. The rationale of conducting this systematic literature review was to highlight the existing research gaps in a particular area of deep learning methodologies and provide useful information on the impact of vegetation indices and environmental factors on crop yield prediction. This systematic literature review has provided various deep learning approaches, features, and factors adopted for crop yield prediction. All the studies were carried out on different types of crops, geological positions, and various features. Overall, the performance and accuracy of the deep learning approach for crop yield prediction are better when compared to traditional machine learning approaches. The deep learning approaches are all equally capable in crop yield prediction based on the factors/parameters used in the model. However, the most effective deep learning approaches for crop yield prediction are the CNN- and LSTM-based approaches. CNN has the ability to find important features that can influence the crop yield prediction. Moreover, LSTM does not only identify the data's variation pattern, but also the time-series data's dependent connection [23]. Based on this study, it is observed that the vegetation indices and the meteorological data are the most used features, where the vegetation indices explain the crops' characteristics and the meteorological data help to monitor the climatic conditions, which has a direct influence on crop yield prediction. It can also be seen that the factors influencing crop yield prediction are selected based on the crop yield and its correlation with other factors. It is also shown that the main challenges of using deep learning approaches and remote sensing for crop yield prediction are in how to improve the working model for better accuracy, the practical implications of the model for providing accurate information about crop yield to agriculturalists, growers, and policymakers, and the issue concerning the black box property.

In future work, we will consider conducting a systematic literature review on articles that use data from hyperspectral sensors for crop yield prediction. Moreover, we will include the specific food crops and classify the techniques and approaches used for different crop yield predictions.

**Author Contributions:** Conceptualization, P.M.; methodology, P.M., S.W. and S.G.; validation, N.H.S. and N.I.; writing—original draft preparation, P.M.; writing—review and editing, S.W., S.G., N.H.S. and N.I.; supervision, S.W., S.G., N.H.S. and N.I. All authors have read and agreed to the published version of the manuscript.

**Funding:** This research received no external funding.

**Informed Consent Statement:** Not applicable.

**Conflicts of Interest:** The authors declare no conflict of interest.

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
