# Peer review of "A Systematic Literature Review on Crop Yield Prediction with Deep Learning and Remote Sensing"

_remotesensing, doi:10.3390/rs14091990_

Round 1

Reviewer 1 Report

The authors have presented a manuscript, which evaluated a systematic review on crop yield prediction with deep learning and remote sensing. Following, I have included some comments aimed to enhance the paper:

·    I think that the authors can improve the format of results demonstration. The authors can highlight better the importance of the results obtained. ·    I see the article very poorly edited, it has a structure in which the table of publications occupies 12 pages of the article out of 33 pages. ·    The authors would have to work on a better presentation and especially on a more in-depth study of all the articles to analyze and discuss them. ·    Consider extending the conclusions and adding a Future works paragraph. The summary and Conclusions, it is better to combine them in only section of conclusions.  

Finally, the topic of this manuscript is interesting; but authors must improve the presentation of their results and discussion.

Author Response

Please see the attachment for the responses. Thank you.

Point 1: I think that the authors can improve the format of results demonstration. The authors can highlight better the importance of the results obtained. I see the article very poorly edited, it has a structure in which the table of publications occupies 12 pages of the article out of 33 pages.

Response 1: Thank you for your comment. Table 2 is now rewritten to summarize the key findings.

Point 2: The authors would have to work on a better presentation and especially on a more in-depth study of all the articles to analyze and discuss them.

Response 2: The paper has been proofread to ensure better presentation and readability. Several paragraphs are also added on page 16 to provide more in-depth study of the articles. 

Point 3: Consider extending the conclusions and adding a Future works paragraph. The summary and Conclusions, it is better to combine them in only section of conclusions.

Response 3: Several sentences are added in the Conclusion section. A paragraph is also added for Future works of this research area.

Point 4: Finally, the topic of this manuscript is interesting; but authors must improve the presentation of their results and discussion.

Response 4: The presentation of the results and discussion have been improved.

Reviewer 2 Report

The paper presents a systematic literature review to synthesize the deep learning approaches, remote sensing technologies, and features that influence crop yield prediction. The article is well written, and the methodology is consistent.  I have suggestions and comments:

1) In the section "2.3.

a) Procedure for article search", the authors describe the systematic review protocol, but they should clarify the specific string they used. They mentioned several synonyms, but the particular string they used is not clear.

b) Regarding the exclusion criteria "Publications that have no-open access". Just 51 papers have achieved as results. In general, the research institutions have access to bases. Maybe it was interesting to remove this criterion to expand the results. The authors affirm: "However, the number of 468 research articles using remote sensing and deep learning approaches for crop yield prediction is still relatively low. "That criteria can limit this conclusion.

2) Concerning the explanation of deep learning techniques, an interesting discussion is about which techniques perform well using small or big databases (generated by remote sensing, for example). Another discussion should involve which papers found in their systematic review have used overfitting or any technique for missing data. Besides, they can also analyze which preprocessing techniques are used to prepare the data before applying the deep learning approaches.

3) The RQ1: the authors should discuss the performance of deep learning techniques. Based on the papers, can the authors differ and recommend which techniques are better applied (involving data or techniques characteristics, deep learning results, for example)?

Author Response

Please see the attachment for the responses. Thank you.

Point 1: In the section 2.3, procedure for article search", the authors describe the systematic review protocol, but they should clarify the specific string they used. They mentioned several synonyms, but the particular string they used is not clear.

Response 1: Thank you for your comment. This is now addressed on page 3.

Point 2: Regarding the exclusion criteria "Publications that have no-open access". Just 51 papers have achieved as results. In general, the research institutions have access to bases. Maybe it was interesting to remove this criterion to expand the results. The authors affirm: "However, the number of 468 research articles using remote sensing and deep learning approaches for crop yield prediction is still relatively low. "That criteria can limit this conclusion.

Response 2:  The publications without access to institution databases were excluded but now we have included Web of Science database for obtaining more relevant articles on crop yield prediction. Table 1 on page 4 is now updated to show the distribution of articles among databases .

Point 3: Concerning the explanation of deep learning techniques, an interesting discussion is about which techniques perform well using small or big databases (generated by remote sensing, for example). Another discussion should involve which papers found in their systematic review have used overfitting or any technique for missing data. Besides, they can also analyze which preprocessing techniques are used to prepare the data before applying the deep learning approaches.

Response 3:  Several paragraphs are now added on page 16 for discussing about the deep learning approaches as well as overfitting, processing and pre-processing of data.

Point 4: The RQ1: the authors should discuss the performance of deep learning techniques. Based on the papers, can the authors differ and recommend which techniques are better applied (involving data or techniques characteristics, deep learning results, for example)?

Response 4:  In Section 4 under RQ1 on page 16, we have included a paragraph for discussing the performance of deep learning approaches.

Reviewer 3 Report

This manuscript deals with a systematic literature review to understand the current literature on the application of deep learning approaches in crop yield prediction using remote sensing. The main goal is to provide a new perspective of research and enable to understand the most recent stage of technological development.

The research is based on these question:

What deep learning approaches are used for crop yield prediction?

What remote sensing technologies are used with deep learning approaches for 86 crop yield prediction?

What are the vegetation indices and environmental parameters used in crop 91 yield prediction?

What are the challenges in using deep learning approaches and remote sensing 95 for crop yield prediction?

I would like to state that these research questions are sufficiently complex for the research.

Section 2  describes the methodology of reearch and article selection criteria (I really appreciate schemes and tables in this section).

Section 3 presents an overview of existing metods from deep learning to remote sensing. I appreciate a breif description of each method here.

Section 4 formulates results of the study. At the beginning there is a figure of article distribution in years, then the summary of of found articles is introduced. The final part of this section is focused on detailed answers to research questions.

The manusript represents a complex research on crop yeald prediction using deep learning and remote sensing. The manuscript is clearly written and forms a very good bases for future work. Owing this fact I recommend to accept the manuscript in present form.--

Author Response

Point 1: The manuscript represents a complex research on crop yield prediction using deep learning and remote sensing. The manuscript is clearly written and forms a very good bases for future work. Owing this fact I recommend to accept the manuscript in present form.

Response 1: Thank you very much for accepting the manuscript.

Reviewer 4 Report

The ms remotesensing-1638514 with the title of A systematic review on crop yield prediction with deep learning and remote sensing; the authors performed a comprehensive systematic literature review to extract and synthesize the deep learning approaches, remote sensing technologies, and features that influence crop yield prediction.

The ms should be significantly improved to be suitable for publication in such high quality journals. In addition, the Figures have to be improved in terms of presentation and quality.

The authors should also improve the title of figures and tables and make them detailed.

L24-26 Who said this? Please cite this relevant ref https://doi.org/10.3390/land10121375

I was looking to see a focus one the most important food crops, and then the authors can go deep for specific crops such as wheat, rice and maize etc.

Table 1. Distribution of articles among various databases, # of articles after removing repeated articles for google scholar is the highest one even higher than Scopus and Science Direct! But, this can affect negatively your analysis and conclusion! Where is the articles from Web of Science, since it considered one of the strongest databases?

I recommend the authors to cite uncited references if they were not their own! This is very important note.

I also recommend the authors to make some statistics for their findings if possible, this will make your ms stronger than the current version. Please, try to make it deeply written, particularly the discussion section.

Good luck,

Author Response

Please see the attachment for the responses. Thank you.

Point 1: The authors should also improve the title of figures and tables and make them detailed.

Response 1: Thank you for your valuable comment. The title for figures and tables are now improved.

Point 2: L24-26 Who said this? Please cite this relevant ref https://doi.org/10.3390/land10121375

Response 2: References are added for lines 24-26.

Point 3: I was looking to see a focus one the most important food crops, and then the authors can go deep for specific crops such as wheat, rice and maize etc.

Response 3: Thank you for your suggestion. The purpose of this literature review is to investigate the use of remote sensing and deep learning in crop yield prediction. Hence the detailed review on specific crop is not included in this paper. In the future research, we  will focus on the specific food crops and the techniques used for the crop yield prediction.

Point 4: Table 1. Distribution of articles among various databases, # of articles after removing repeated articles for google scholar is the highest one even higher than Scopus and Science Direct! But, this can affect negatively your analysis and conclusion! Where is the articles from Web of Science, since it considered one of the strongest databases?

Response 4: Based on your suggestion, a search is conducted on using Web of Science database. The total number of articles retrieved from Web of Science is included in Table 1. Several relevant articles obtained from this database are also discussed in Table 2.

Point 5: I recommend the authors to cite uncited references if they were not their own! This is very important note.

Response 5: The manuscript has been thoroughly checked and relevant references are now added.

Point 6: I also recommend the authors to make some statistics for their findings if possible, this will make your ms stronger than the current version. Please, try to make it deeply written, particularly the discussion section.

Response 6: A paragraph discussing overfitting, processing and pre-processing of data is included in Section 4 under RQ1 on page 16. Also, statistical representation of some findings in the form of Figure 4 and 5 are included in Section 4 under RQ2 on page 19.

Round 2

Reviewer 1 Report

The authors have presented a manuscript, which evaluated a systematic review on crop yield prediction with deep learning and remote sensing. Following, I have included some comments aimed to enhance the paper:

·    I think that the authors can improve the format of results demonstration. The authors can highlight better the importance of the results obtained. ·    I see the article very poorly edited, it has a structure in which the table of publications occupies five pages of the article out of 30 pages. I think is very poor.

·    I see the features that influence crop yield are grouped in the 65 until 77 (page 20, 21 and 22) is very poor edited in this article. ·    The authors would have to work on a better presentation and especially on a more in-depth study of all the articles to analyze and discuss them. ·    Consider extending the conclusions and adding a Future works paragraph. The summary and Conclusions, it is better to combine them in only section of conclusions.  

Finally, the topic of this manuscript is interesting; but authors must improve the presentation of their results and discussion.

Author Response

Point 1: I think that the authors can improve the format of results demonstration. The authors can highlight better the importance of the results obtained. ·    I see the article very poorly edited, it has a structure in which the table of publications occupies five pages of the article out of 30 pages. I think is very poor.

Response 1: Thank you for your valuable comment. Table 2 has been revised to summarize the key points and discussion on Table 2 is presented on page 10. 

Point 2:  I see the features that influence crop yield are grouped in the 65 until 77 (page 20, 21 and 22) is very poor edited in this article. ·    The authors would have to work on a better presentation and especially on a more in-depth study of all the articles to analyze and discuss them. ·    Consider extending the conclusions and adding a Future works paragraph. The summary and Conclusions, it is better to combine them in only section of conclusions.  

Response 2: Table 3 is now added to summarize the features that influence crop yield. The conclusion section has been rewritten and future work paragraph is added.

Point 3: Finally, the topic of this manuscript is interesting; but authors must improve the presentation of their results and discussion.

Response 3: The presentation of the results and discussion has been improved.

Reviewer 4 Report

The authors revised the ms according to my suggestions, thanks

Author Response

Thank you for accepting the paper.

Round 3

Reviewer 1 Report

The authors have presented a manuscript, which evaluated a systematic review on crop yield prediction with deep learning and remote sensing:

·   I think that the author’s haven improved the format of results demonstration.   ·   The author’s haven worked on a better presentation. ·   I think is well for to publication. 

Finally, the topic of this manuscript is interesting.

Author Response

Thank you for accepting the paper.